# Sustaining indigenous *Maasai Alalili* silvo-pastoral conservation systems for improved community livelihood and biodiversity conservation in East African rangelands

**Elkana Hezron** *, Issakwisa B. Ngondya, Linus K. Munishi

Department of Sustainable Agriculture, Biodiversity and Ecosystem Management, School of Life Sciences and Bioengineering, The Nelson Mandela African Institution of Science and Technology, Arusha, Tanzania

* elkana.hezron@nm-aist.ac.tz

## Abstract

*Alalili* system is one among the fewest remnant African indigenous and local knowledge systems that is traditionally practiced by *Maasai* pastoral communities to conserve certain portions of rangeland resources such as pastures and water for subsequent grazing during dry seasons. Despite its existence, East African rangelands face diverse threats from tenure security, unsustainable practices, climate, and land-use change that are notably endangering the biodiversity, livelihoods, and ecosystems in the landscape. Like other indigenous conservation systems, the sustainability of *Alalili* systems is being threatened, as *Maasai* communities are in transition due to continuous socio-cultural transformations coupled with increased livestock and human populations. We aimed to capture and document the existing occurrence and potential of *Alalili* systems as a pathway to improve resilience and sustain both biodiversity conservation and community livelihoods in rangeland areas of northern Tanzania. A cross-sectional research design was applied with the adoption of both purposive and stratified random sampling techniques to distinctively characterize the *Alalili* systems by land use and tenure types. Our results identified the existence of both communal and private *Alalili* systems. Their sizes varied significantly across types (t = 4.4646, *p < 0.001*) and land uses (F = 3.806, df = 3, *p = 0.0123*). While many (82%) of these *Alalili* systems are found in the communal land, our observations show a re-practice of *Alalili* systems in the private land is considered largely a re-emerging strategy for securing pastures in the face of local and global change. More than half (73%) of *Alalili* systems were found within game-controlled areas with little representation (about 8%) in non-protected land. Therefore, their sustainability is threatened by anthropogenic and climatic pressures, making their persistence more vulnerable to extinction. We recommend mainstreaming these practices into core pasture production and management areas, facilitating their reinforcement into policy and practices.

**Data Availability Statement:** All relevant data are within the manuscript and its Supporting Information files.

**Funding:** This work was funded by the Nelson Mandela African Institution of Science and Technology through African Development Bank (AfDB) (Grant No: P-Z1-IA0-016), Higher Education for Economic Transformation (HEET) project (Grant No: IDA68870) as well as the Rufford small research grants (Grant No: 37388-1). The funders had no role in study design, data collection and analysis, decision to publish, or preparation of the manuscript.

**Competing interests:** The authors have declared that no competing interests exist.

# 1. Introduction

*Alalili* is a traditional silvo-pastoral conservation system indigenous to *Maasai* communities through which certain portions of rangelands are conserved during the wet season for improved natural regeneration of vegetative biomass useful for grazing during dry seasons [1–6]. Indigenous and Local Knowledge (ILK) including traditional silvo-pastoral conservation systems, is important for promoting sustainable use of resources, supporting resilience, and preventing further damage to the natural world [7, 8]. This knowledge has been part of the co-evolutionary relationship with people, biodiversity, and the environment through enhanced carbon sequestration and storage, wildlife habitat, increasing livestock productivity, and strengthening rural livelihoods [6, 7]. Despite its vital role, this cultural component is missed in scientific studies, the absence of which may make it vanish in silence, jeopardizing biodiversity conservation and indigenous peoples' livelihoods [9]. *Alalili* is one such traditional conservation system that is practiced by *Maasai* communities across East Africa to promote biodiversity conservation and sustainable pasture management in rangelands [1, 4–6]. Having been practiced since the arrival of *Maasai* communities in East African rangeland around the 18th century [10–14], *Alalili* systems are currently regarded as useful traditional rangeland management initiatives [13, 15–17]. They are undertaken as alternative measures to alleviate the effects of declined pastoral mobility caused by land-use and land-cover changes (LULCC), population increase, and climate change [5, 18, 19]. They can play important roles in the phase of global changes by promoting resilience in community livelihood and biodiversity conservation which are significant economic dimensions for pastoral and agro-pastoral communities. They are useful fodder resources for both livestock and wildlife, potential pollinator conservation areas, climate change mitigation sites through carbon sequestration, and nature-based restoration solutions to degraded ecosystems [17, 20]. For instance, similar communal rangeland management systems, alternatively referred to as *Maasai Olopololi*, have been supportive of restoring the degraded grasslands during the regreening campaign of the Amboseli ecosystems in southern Kenya [21, 22]. They integrate human, livestock, and wildlife through the provisioning of useful habitats and breeding sites to thousands of organisms that depend on *Alalili* systems for their survival [11, 18]. They provide assurance to the endurance of wild space which maintains high levels of biodiversity while adding value to rangeland resilience over local climate change [15, 20, 23].

Having a close association with *Alalili*, the rangelands of northern Tanzania and southern parts of Kenya are significantly known to support millions of biota communities inclusive of the human population [5, 24, 25]. Community reliance on rangelands has been evident from the global, regional, and local perspectives whereby its average terrestrial landscape coverage is approximated to be 47% globally, 36% in Asia, 30% in Africa, and 74% of the total land in Tanzania [8, 25]. Although, recent studies report that rangelands are prone to global changes coupled with declined feed production, impacts of climate change, and loss of resistance as well as reduced sustainability [8, 26]. LULCC has led to intense environmental and socio-economic impacts that disrupt sustainable land use practices including rangelands leading to insufficient pastures [10, 27–31]. Pastoral communities adapted the traditional pasture conservation practices as an alternative measure for restoring and strengthening the health of rangelands to sustain their productivity and livelihoods [1, 31–35]. In Asia particularly India, the system known as "*sacred groves*" is preserved for the purpose of social-cultural and religious practice [33, 36–38]. In the Middle East on the other hand they practice *imā* as a reserved pasture, whereby trees and grazing lands are secured from indiscriminate harvest on a temporary or permanent basis [39]. In Africa, *Kalo* is a silvo-pastoral conservation system practiced by the Borana communities of Southern Ethiopia [40]. In Tanzania, *Ngitili* is practiced by Sukuma pastoralists,

*Milanga* is practiced by Gogo, *Radanenda* is practiced by Barbaig while *Alalili* is practiced by *Maasai* pastoral communities [5, 16, 23, 41, 42].

Although the indigenous *Maasai Alalili* silvo-pastoral conservation systems have been successful in the past depicting its long-term historical benefits within rangelands and ecosystem resources, its practice as a resilience back-up to rangeland degradation is not fully optimized [2, 3, 22]. The reduction in its practices is exacerbated by increased human and livestock population, socio-cultural transformations, changes in land tenure, and unpredictable land use and climate changes [8, 43, 44]. In addition, these socio-cultural changes and extreme events have accelerated the degradation of rangelands, making biodiversity, livelihoods, and ecosystems more threatened; this has pushed for more proactive measures in response to these pressures including relocating vulnerable pastoral communities in the Ngorongoro district to a much less degraded but more productive rangeland for livestock and settlement [12, 43]. Despite the importance of the *Maasai Alalili* silvo-pastoral conservation and management systems, their little recognition in literature and under-representation in the existing policies and regulatory frameworks might undermine their utility and importance in rangeland management practices, thus, threatening their value and sustainability in resilience management [1, 5, 17]. Additionally, *Alalili* practices are in the blink of extinction, as *Maasai* societies in most areas are in transition with respect to continuous socio-cultural transformation [17, 45–47]. A proven misconnection in the transmission of this traditional heritage from elders to the younger generation, who are gradually shifting to other economic activities in cities such as decorative saloons and security guards, exposes *Alalili* to encroachment threats [3, 5, 12]. The indeterminate decrease in size and abundance of *Alalili* systems, depicting their possible decline across the *Maasai* rangeland areas of northern Tanzania and southern Kenya, has been recently reported by rangeland ecosystem management stakeholders [22, 44, 48]. In the face of this, there is a need to have them documented and understand their current status that can be used as a potential pathway to promote both sustainable conservation of biodiversity and community resilience and livelihood development across *Maasai* land. This study provides information on the status of types, size, and distribution of *Alalili* in different land use systems across the rangeland of northern Tanzania. It provides useful recommendations utilized as a means to guide communities and other stakeholders toward the preservation and practice of this important heritage and how these can be sustained in the rapidly changing and uncertain world of the Anthropocene.

Therefore, being conducted for the first time, this study aimed to capture and document the potential *Alalili* silvo-pastoral conservation systems available in rangelands located within or adjacent to different land uses of northern Tanzania. Also, the study explored the livestock herd size featuring the surveyed *Alalili* systems and its relationship across size and age of *Alalili* systems to initiate future studies in the determination of biomass and stocking rate. This was achieved so that they can be promoted as a pathway to improve the resilience of pastures and sustain both biodiversity conservation and community livelihoods in rangeland areas.

## 2. Methods

### 2.1 Description of the study area

The study was conducted in *Maasai* pastoral society across the regions of northern Tanzania, located between 2˚12'04" and 5˚56'29" South and 36˚11'43" and 36˚51'30" East covering five sampled Districts. These included Longido, Monduli, and Ngorongoro districts for the Arusha region as well as Simanjiro and Kiteto districts for the Manyara region (Fig 1). The area is within the East African rift system, and its landscape lies at an elevation ranging between 659 and 2,123 m above sea level characterized by the highest mean monthly temperature of 33˚C,

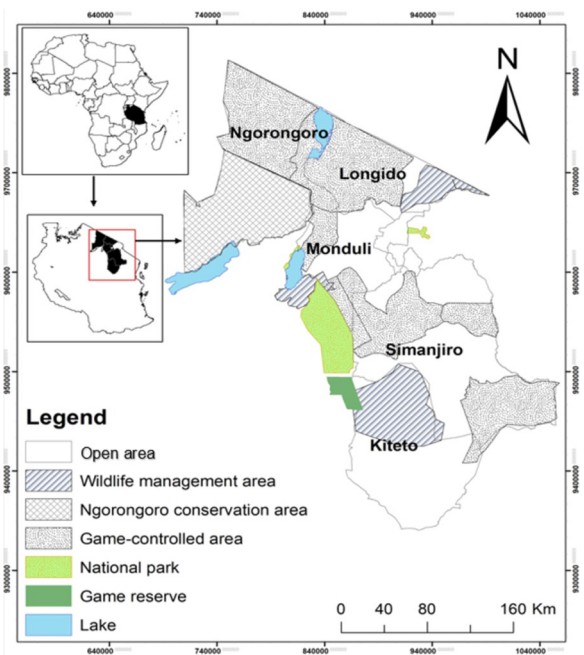

**Fig 1. A map of the study area portraying the surveyed land uses of northern Tanzania rangeland.**

while the lowest mean temperature is 13˚C [15]. The mean annual rainfall ranges between 450 and 1,200 mm [49]. The region is featured with bimodal rainy seasons, with the short season falling between October and December, while the long rainy season falls between March and May [50, 51]. In normal years, the rainy season stretches from January to April [47, 52, 53]. The vegetation cover in the region ranges from grassland *Maasai* steppe to bushland in the lowlands and upland tropical thick forests growing on well-drained soils of fertile land, which is alternatively used for agriculture, mining activities, grazing, and game-controlled areas [47, 50, 54]. The vegetation cover in the low grassland and midlands of the region generally includes scattered trees and shrubs dominated by *Acacia* spp., *Combretum* spp., and *Brachystegia* spp. as well as grasses in the genus *Cenchrus* spp. and *Cynodon* spp. The dominant communities of the *Maasai* tribe in the villages are engaged in social-economic activities such as livestock keeping, agriculture, mining, jewelry, and beekeeping. Livestock keeping is an important and main economic activity that forms part of family wealth and savings [1, 35, 55].

## 2.2 Categorization of land uses and tenure types in northern Tanzania

The Land Act of Tanzania (enacted in 1999) clarifies that "all land in Tanzania is public and is vested in the President as a trustee on behalf of all citizens" [56]. The law under the Ministry of Lands, Housing, and Housing Development is in charge of facilitating an equitable distribution of and access to land by all citizens [56]. Therefore, the Land Act Cap. 113 classified the Tanzanian land tenure into three main categories; (a) general land, (b) village land, and (c) reserved land. The general land refers to all public land with the exception of village land (including unoccupied open areas or unused village land) and reserved land. Village land refers to any part of the general land other than reserved land transferred to a village [57]. Its establishment is aimed at providing access of land to indigenous people living within and around reserves for settlement, agriculture (crop production and pastoralism), and infrastructure (education, health, business, transport, and the related facilities). This is done with regard

to the Local Government (District Authorities) Act Cap. 287 and the Land Tenure (Village Settlements) Act No. 27 of 1965 [58]. Reserved land refers to the land that is purposely utilized for fortification and conservation of biodiversity and its associated resources which are profoundly controlled by the State [56, 59]. These are comprised of strictly protected areas (PAs) such as national parks and game reserves (forestry and nature reserves) [60]. They are also comprised of non-strictly PAs such as the Ngorongoro conservation area, game-controlled areas, game-open areas, and wildlife management areas as defined in the Wildlife Conservation Act of 1974[Cap. 283] and wildlife policy endorsed in 1988 [60, 61].

National parks and game reserves are defined as strictly protected areas whereby no activities other than wildlife conservation and game viewing are allowed [59, 62]. Ngorongoro Conservation Area (NCA) is an area dedicated to multiple land uses including *Maasai* culture and tradition conservation, livestock grazing by pastoral communities, and temporary residence thus forming a peaceful co-existence with wildlife [56, 59]. However, in NCA villages are not permanently allowed to own the land, but rather, land tenure is generally devoted to the managing authority; the NCAA [61, 63]. A game-controlled area (GCA) refers to a type of protected area whereby a village's land and resource use other than wildlife conservation is restricted by the law [59, 64]. Activities such as settlements, crop cultivation, and mining are restricted while livestock grazing and wildlife hunting can only be acceptable under license or written permission of the managing authority [59]. Wildlife management area (WMA) refers to an area established with regard to initiatives of implementing a community-based conservation of natural resources [59, 65]. The gazettement of WMA is done with the main purpose of securing wildlife buffer areas outside core protected areas, which are used by local community members and within the village land [59, 65]. In contrast to GCA, WMA comprises the portion of a village land through which communities have equitable access to the distribution of benefits from enterprises aimed at promoting wildlife conservation, enhancing economic development, and poverty reduction [56, 59, 65].

Therefore, being conducted in *Alalili* systems that form part of rangelands in northern Tanzania, the study covered two main land uses, i.e., protected areas vested to reserved land (comprised of WMA, GCA, NCA, national parks, and game reserves) and unprotected open areas vested to village land [66]. It considered *Alalili* systems located within or adjacent to reserved land and unprotected open areas predominantly utilized for pastoralism. For this study, unprotected open areas under the occupation of village lands will be termed "open areas" from now onwards. It is from these land use categories, the northern Tanzania regions for this study were clustered and mapped to capture *Alalili* systems that are practiced for grazing activities by the *Maasai* pastoral communities. The existence of *Alalili* systems in rangelands within studied land uses was approved by the government authorities.

## 2.3 Ethics statement

Access to *Alalili* systems in the studied land use categories received an approval from the Tanzanian Commission for Science and Technology (COSTECH) (permit number 2022-222-NA-2022-005 dated 20[th] April, 2022), Ministry of Natural Resources and Tourism through Tanzania Wildlife Research Institute (TAWIRI) (permit number TWRI/RS/22"G"/21 dated 22[nd] April 2022), Tanzania Wildlife Management Authority (TAWA) (permit number AE.542/712/01 dated 10[th] May 2022), and the Ngorongoro Conservation Area Authority (NCAA) (permit number BD.158/711/01-B/100 dated 09[th] May 2022). Permission for research in communities was obtained from relevant local and district authorities.

## 2.4 Research design and sampling techniques

Prior to main data collection, a reconnaissance survey was conducted to confirm the availability of *Maasai Alalili* systems within districts in the proposed study regions. The sampling in this study involved a cross-sectional research design whereby both purposive and stratified random sampling techniques were applied. Administrative boundaries of the Arusha and Manyara regions were marked through Quantum Geographical Information System (Q-GIS) software version 3.12.3 [67] to enhance the sampling process of the *Maasai* land with the assistance of shape files collected from Tanzania Wildlife Research Institute (TAWIRI). Five districts from the two regions were sampled purposively, targeting pastoral and agro-pastoral *Maasai* communities (Fig 1). A purposive sampling of the five districts dominated by the *Maasai* communities was adopted to enhance the identification process of the *Alalili* systems and livestock herd size grazed in the sampled *Alalili* systems through key informant interviews and focused group discussion respectively [68, 69]. Key informant interviews were conducted in each district for the purpose of identifying villages that have *Alalili* systems as well as preparing the list of all *Alalili* systems from identified villages. Key informants included district game officers (DGOs), rangeland management officers (ROs), ward/village executive officers (VEO/WEO), and members of the village rangeland management committee. A total of 61 interviews were done in the study area whereby five were conducted at district levels and 56 at ward levels. DGOs and ROs from all five districts were consulted through direct physical meetings conducted at each district council office headquarters during a reconnaissance survey. WEO/VEO from a total of 88 wards of the sampled districts were contacted with assistance from DGOs and ROs via phone/mobile calls considering the limited time, available resources, and broader range of the study area. The interviews between DGOs, ROs, and WEO/VEO enhanced to reach out the leaders of the respective village rangeland management committees who gave the list of all *Alalili* systems present in their villages. A total of 298 *Alalili* systems were identified and numbered continuously, from which 40% (119 *Alalili* systems) were sampled to cover a wider range of the study area through randomization technique with the adoption of random number tables. Focused group discussion comprised of one member of the village rangeland management committee and two or three members of the pastoral households who graze livestock in the sampled *Alalili* systems for estimation of livestock herd size.

Stratification of the study area was conducted with regard to the existing land use practices in the Arusha and Manyara regions. Six types of land use strata were plotted through Q-GIS and distinguished using the topographic map of sampled regions. The land use strata were open areas, WMA, GCA, NCA, national parks, and game reserves. National parks and game reserves were not accounted for mapping of *Alalili* as *Maasai Alalili* silvo-pastoral conservation systems are not allowed in the national parks and game reserves. Stratification of the sampled villages that comprised the sampled *Alalili* systems was done based on different land-use systems with respect to their distribution. An empirical evidence-based method (Ground-truthing technique) was applied through field visitation and direct observation. DGOs and ROs assisted in verifying the exact land use types in the study area whereby GCA covered the largest proportion (45%) followed by NCA (20%), open areas (19%), and WMA (16%) [70].

## 2.5 Data collection

All *Alalili* samples were visited in each sample village and their respective locations were recorded at the center of each surveyed *Alalili* system with a GPS to enhance mapping through the Q-GIS platform. Since the specific boundaries and actual sizes (ha) of each *Alalili* were not known, four radii were recorded from the center of each visited *Alalili* system to the north, south, east, and west directions assuming that *Alalili* has an irregular polygon. The *Alalili*

boundaries at each end of the four directions were demarcated with a help from the local field assistant for each specific village. The common local trails within the surveyed *Alalili* systems were considered as transects from which the radius was measured from the center to every end. A handheld GPS was used to measure the radius where we managed to walk toward the boundaries at each end for small-sized *Alalili* systems, while the boundaries of the large-sized *Alalili* systems were reached out by a field vehicle. The four radii were recorded in the field data sheet from which an average radius of each *Alalili* system was obtained. The radius was used to compute the actual ground area of a specific *Alalili* system (Eq 1). Moreover, Google background image in the Q-GIS environment was used to locate the extent in terms of sizes for each *Alalili* through digitization. The digitization of each *Alalili* polygon was done using Google Earth Pro with the image captured by Landsat/Copernicus from the baseline landcover maps taken between February 2018 to December 2020 [71]. The local field assistant helped to identify the boundaries of each *Alalili* in the zoomed satellite view on the screen. The two values were compared and the average was taken into consideration to decide the actual area covered by each *Alalili* system (Eq 2).

$$A = \pi r^2 \qquad \text{(Eq 1)}$$

Whereby, A is the actual ground area of *Alalili* (km$^2$) and r is an average radius of *Alalili* (km)

Estimated Area of *Alalili* (ha) $=$ (Actual ground area (ha) $+$ Digitized area (ha))$/2$ (Eq 2)

Data were collected from late April to early December 2022 whereby the types of *Alalili* systems (communal and private *Alalili*) were identified through participatory field visits and semi-structured interviews (SSI). The participatory field visits involved the research team comprised of a research team leader, research assistant, DGO/RO, a leader, and other two members of the village rangeland management committee (local field assistants). The leader of the village rangeland management committee helped to identify the type of the visited *Alalili* systems. SSI through focused group discussion with DGO/RO and the local field assistants, that adopted the guiding questionnaires and checklist [72], was used to determine other descriptive features of *Alalili* systems. This includes the approximated livestock herd size, livestock categories accommodated by each surveyed *Alalili* system, age, approximated distance to water sources, management modality, and allocated time to open the *Alalili* for grazing.

## 2.6 Data analysis

Mapping of the spatial distribution of the *Alalili* system across *Alalili* type, size, and land use categories was done in Q-GIS version 3.12.3. Prior to mapping, the area of *Alalili* was categorized into three classes (small, medium, and large). The class sizes were obtained via the standardization method whereby all *Alalili* systems that had an area below the mean range (mean ± SE) were considered small, *Alalili* systems with an area within a mean range were considered medium and those with above mean range were regarded as large. Excel spreadsheet for Windows 2011 was used in data processing, organization, and data sorting to ensure smooth analyses. The abundance and distribution of *Alalili* systems were presented in percentages and tabular forms depicting their proportional dispersion within the mapped rangelands of northern Tanzania and their respective land uses. The chi-square test was used to understand the variation of abundance and distribution across *Alalili* types and land use categories [33, 47]. Independent sample t-test was used to understand the variation of total area covered by *Alalili* systems across types while analysis of variance (ANOVA) was used to understand the variation of total area covered by *Alalili* systems across land use and size categories. Prior to analyzing the effect of land use and types of *Alalili* on the size of *Alalili* systems, the

Shapiro-Wilk test for normality was conducted. A homogeneity test of variance was also done by using Levene's test. A generalized estimating equations (GEE) model was applied to assess the effects of land uses and *Alalili* types on the size of *Alalili* systems (Eq 3). Similarly, the GEE model was used to assess the relationship between *Alalili* size and livestock herd size (Eq 4) through the R version 4.2.3 [73, 74].

$$\text{geeglm(formula} = \text{Area} \sim \text{Land use} + \text{Types of } \textit{Alalili} + \textit{Alalili} \text{class size,}$$
$$\text{family} = \text{gaussian(), data} = \text{e, id} = \text{Types of } \textit{Alalili, corstr} = \text{"exchangeable")} \quad \text{(Eq 3)}$$

$$\text{geeglm(formula} = \text{Total Livestock} \sim \text{Types of } \textit{Alalili} + \text{Land use} + \text{Age} + \text{Area of } \textit{Alalili,}$$
$$\text{family} = \text{gaussian(), data} = \text{A, id} = \text{Types of } \textit{Alalili, corstr} = \text{"exchangeable")} \quad \text{(Eq 4)}$$

The GEE model equation considered the aspects of land use, type of *Alalili*, and size of *Alalili* as factors whereby the baseline variables were GCA, Communal *Alalili*, and Large size respectively. The dependent variables in the model equations comprised of size of *Alalili* systems in terms of area (ha) and livestock herd size (total number of livestock grazed in *Alalili* systems). The independent variables comprised of types of *Alalili* systems, land use categories, class size, and age of surveyed *Alalili* systems. A p-value of $p < 0.05$ was considered significant.

## 3. Results

### 3.1 Classification of *Maasai Alalili* silvo-pastoral conservation systems

Two types of *Alalili* systems; communal and private were identified in the *Maasai* pastoral communities in both Arusha and Manyara regions. Communal *Alalili* were reported to be silvo-pastoral systems owned and managed by a village or a village section for the purpose of providing fodder/forage to livestock during the acute dry season. Further classification of communal *Alalili* systems in terms of size determined three categories; small (1–924 ha), medium (925–1,320 ha), and large (above 1,320 ha). They are generally large in size compared to private *Alalili* and allow large numbers of livestock, weak and/or sick, as well as lactating herds to graze based on the rules and regulations set up by the village rangeland management committee. The livestock are allowed to graze in the communal *Alalili* during the acute dry season only in numbers ranging between 50–450 herds with respect to the size of the communal *Alalili* being grazed. Many of the communal *Alalili* (74%) were found far from the settlement areas, they were found in the wilderness (either on high or low land) and were not fenced (Fig 2). They were also reported to be found in the protected areas where they are utilized as shared grazing resources between livestock and wildlife. The majority of them were old (80 and 90 years) although some were reported to be established recently (10 or 15 years old).

Private *Alalili* on the other hand were reported to be silvo-pastoral conservation systems that are owned and managed by individual families or a clan referred to as a *Boma*. Some private *Alalili* were observed to be privately owned by investors who aimed at high production of livestock products. They are further classified into three size categories; small (1–156 ha), medium (157–261 ha), and large (above 261 ha). They are generally considered smaller in size compared to communal *Alalili* systems to enhance suitability and sustainable fodder availability to only a smaller number of livestock at a range of 30–150 herds. Occasionally, private *Alalili* owned by investors has a large size (250–1,000 ha) considering the large number of livestock being kept by an investor (500–3,000 herds). They are meant to provide fodder to calves, weak and/or sick, as well as lactating livestock. The grazing season allowed to feed such livestock in the private *Alalili* is determined by the *Alalili* owner. It was reported that private *Alalili* have no definite grazing season and the majority of them (94%) were closer to the

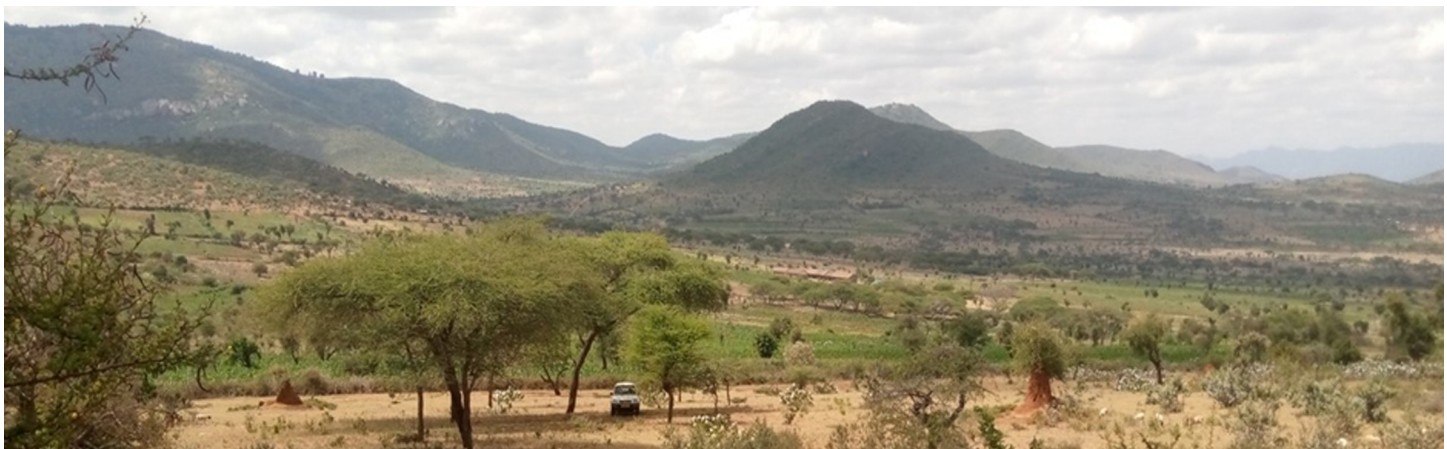

**Fig 2. One among the communal *Maasai Alalili* systems surveyed in northern Tanzania.**

residential area (around the settlement areas). In contrast to communal *Alalili*, private *Alalili* were found far from the wilderness and some of them were confined with a live or local fence for protection against wildlife encroachment and depredation (Fig 3). They are also utilized as shared grazing resources between livestock and wildlife that graze/hide within them during night hours especially when crossing from one PA to another. Most of them are up to 30 years old although some are aged up to 70 years.

## 3.2 Size of *Alalili* systems across types

The size of communal *Alalili* systems varied significantly (t = 4.4646, *p < 0.001*) from private *Alalili* systems. Communal *Alalili* conservation systems had the largest mean area coverage (1,122±198 ha) compared to private *Alalili* (209±52.1 ha) (Fig 4).

Furthermore, there was a significant variation in the size of *Alalili* between GCA and other land use categories (F = 3.806, df = 3, *p = 0.0123*). *Alalili* in GCA had the largest mean area (1409±307 ha) followed by those in WMA (608±224 ha), NCA (480±167 ha), and open areas

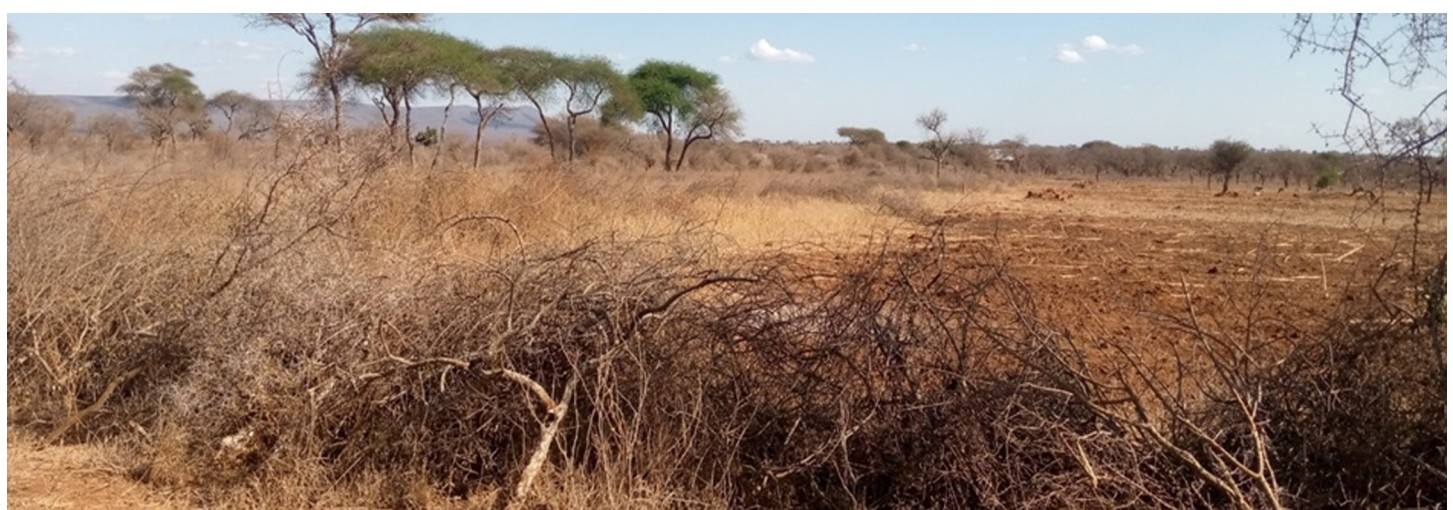

**Fig 3. One among the private *Maasai Alalili* system surveyed in northern Tanzania.**

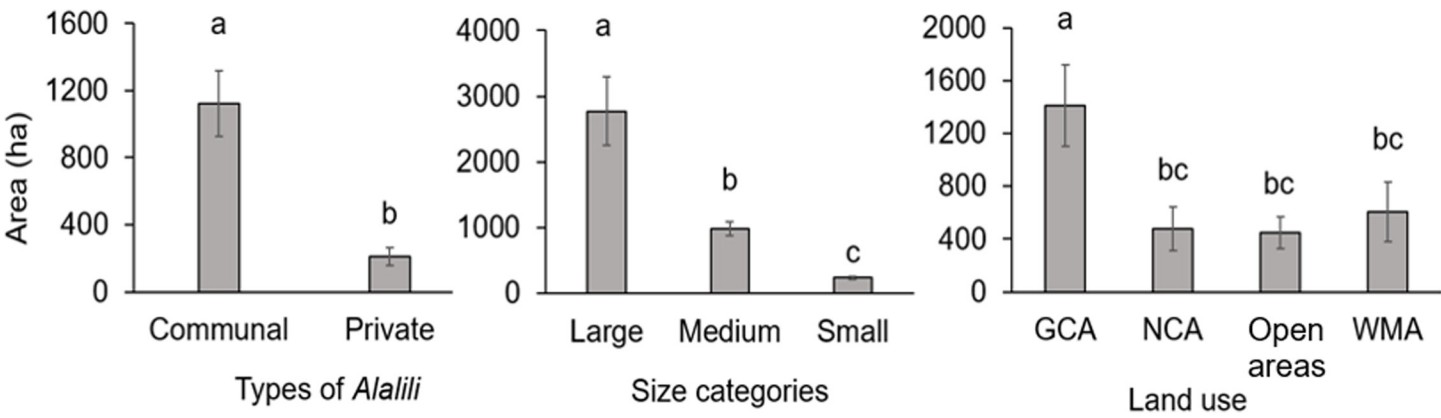

**Fig 4. The size variation of *Alalili* systems across types, size categories, and land uses (the [a, b, c]; depicts mean areas that are significantly different ($p < 0.05$)).**

(446±119 ha). There was a significant variation (F = 27.175, df = 2, *p < 0.001*) in area coverage across size categories of *Alalili* systems whereby large, medium, and small-sized *Alalili* had a mean area of 2773±522 ha, 985±106 ha, and 240±27 ha respectively (Fig 4).

**3.2.1 Overall area coverage of *Alalili* silvo-pastoral conservation systems.** Generally, 119 *Alalili* systems covered an overall area approximated to only 3% of the entire study area (4,087,984 ha). Out of the pre-determined 3%, communal *Alalili* comprised 96% of the total area while private *Alalili* comprised 4% only of the total area. On the other hand, 73% of the overall *Alalili* area was covered by large-sized *Alalili* systems, followed by 16% of the total *Alalili* area covered by small-sized *Alalili* systems while medium-sized covered 11% (Table 1).

**3.2.2 Size of *Alalili* systems within land use categories.** In GCA there was a significant variation in the size of *Alalili* between types (t = 3.7252, *p < 0.001*) whereby communal *Alalili* had a mean area of 1,700±374 ha while private *Alalili* was 271±85 ha (Table 1). In WMA, the size between communal and private *Alalili* systems varied significantly (t = 2.5073, *p = 0.0319*). Communal *Alalili* had a mean area of 964±339 ha while for private *Alalili* was 98.9±62 ha. Although, in the open areas, there was no significant variation in size between communal and private *Alalili* systems (t = 1.8904, *p = 0.0759*) whereby communal *Alalili* had a mean area of

**Table 1. Summary of *Alalili* area coverage across types, land use, and size categories.**

| Land use | Types of *Alalili* | Area by size (ha) | | | Total Area (ha) |
|---|---|---|---|---|---|
| | | Large | Small | Medium | |
| GCA | Communal | 65,207 | 8,016 | 6,658 | 79,880 |
| | Private | 2,763 | 302 | 191 | 3,255 |
| NCA | Communal | 5,087 | 3,921 | 2,514 | 11,522 |
| | Private | 0 | 0 | 0 | 0 |
| Open Areas | Communal | 2,215 | 3,409 | 2,211 | 7,835 |
| | Private | 267 | 127 | 250 | 644 |
| WMA | Communal | 7,197 | 2,446 | 0 | 9,643 |
| | Private | 464 | 229 | 0 | 693 |
| Total | Communal | 79,705 | 17,792 | 11,383 | 108,880 |
| | Private | 3,493 | 657 | 441 | 4592 |
| | Total | 83,199 | 18,449 | 11,824 | 113,472 |

* Abbreviations: GCA, Game Controlled areas; NCA Ngorongoro Conservation Area; WMA, Wildlife Management Area.

490±139 ha while that of private *Alalili* was 215±44 ha. In contrast to other land use categories, NCA comprised of communal *Alalili* systems only with a mean area of 480±167 ha (Fig 5).

**3.2.3 Overall area coverage of *Alalili* systems across land use and sizes.** *Alalili* systems in the GCA had the highest proportion area covering 73% of the total *Alalili* area and it was only 4.5% of the entire area shielded by GCA (1,850,000 ha). NCA had a total *Alalili* area whose proportion was 10%, approximated to only 1.4% of the entire area covered by NCA (809,440 ha). *Alalili* systems in WMA had a total area coverage whose proportion was 9% which was approximated to only 1.6% of the entire area covered by WMA (643,544 ha). The open areas on the other hand had a total *Alalili* area coverage whose proportion was 8% being approximated to only 1.1% of the general area covered by studied village lands (785,000 ha). In the GCA, about 81.8%, 10%, and 8.2% of its total *Alalili* area were covered by large, small, and medium-sized *Alalili* systems respectively. Likewise, in the NCA, about 44.2%, 34.0%, and 21.8% of its total *Alalili* area were covered by large, small, and medium-sized *Alalili* systems respectively. In the WMA, about 74.1% and 25.9% of its total *Alalili* area were covered by large and small-sized *Alalili* respectively while medium-sized *Alalili* systems were not identified. In the open areas, about 41.7%, 29.3%, and 29.0% of its total *Alalili* area were covered by small, large, and medium-sized *Alalili* systems respectively (Table 1).

**3.2.4 Area coverage of communal *Alalili* systems across land use and size.** Communal *Alalili* systems in GCA had the highest area covering 73% of the total communal *Alalili* area. NCA was the second land use having communal *Alalili* systems with an area covering 11% of the total communal *Alalili* area. Communal *Alalili* systems in WMA covered an area equivalent to 9% of the total communal *Alalili* area while *Alalili* systems in the open areas covered an area equivalent to 7% of the total communal *Alalili* area (Table 1). Considering the area of communal *Alalili* systems across sizes within land uses, it was found that in GCA, about 81.6%, 10.0%, and 8.4% of its total communal *Alalili* area were covered by large, small, and medium-sized *Alalili* respectively. In NCA, about 44.2%, 34.0%, and 21.8% of its total communal *Alalili* area were covered by large, small, and medium-sized *Alalili* respectively. In the WMA on the other hand, about 74.6% and 25.4% of its total communal *Alalili* area were

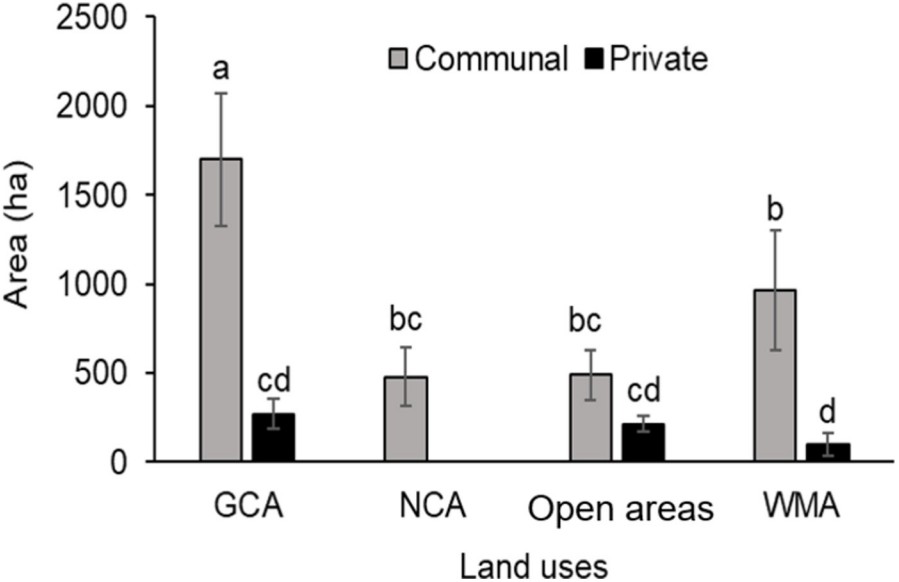

**Fig 5. The variation in sizes of *Alalili* types within the surveyed land use categories (the [a, b, c]; depicts mean areas that are significantly different ($p < 0.05$)).**

covered by large and small-sized *Alalili* systems respectively in the absence of medium size. In the open areas, about 43.5%, 28.3%, and 28.2% of its total communal *Alalili* area were covered by small, large, and medium-sized *Alalili* systems respectively (Table 1).

**3.2.5 Area coverage of private *Alalili* systems across land use.**   GCA had the highest area coverage of private *Alalili* systems with a total proportion of 71% of the total private *Alalili* area. WMA had private *Alalili* systems with an area covering 15% of the total private *Alalili* area while those in the open areas covered 14% of the total private *Alalili* area. On the other hand, no private *Alalili* systems were found in NCA. Considering the area of private *Alalili* systems across sizes within land uses, it was found that in GCA, about 84.9%, 9.3%, and 5.9% of its total private *Alalili* area were covered by large, small, and medium-sized *Alalili* systems respectively. In the WMA, about 67% and 33% of its total private *Alalili* area were covered by large and small-sized *Alalili* systems respectively and no medium-sized *Alalili* systems were observed. In the open areas, about 41.5%, 38.8%, and 19.7% of its total private *Alalili* area were covered by large, medium, and small-sized *Alalili* systems respectively (Table 1).

## 3.3 Overall abundance and distribution of *Alalili* enclosures

A total of 119 individual *Alalili* silvo-pastoral conservation systems were recorded in the study area. The identified traditional systems generally showed a spatial distribution across the study area (Fig 6).

**3.3.1 Overall abundance and distribution across types of *Alalili*.**   There was a significant variation ($\chi^2 = 47.3$, df = 1, $p < 0.001$) of abundance between communal and private *Alalili* systems. The highest abundance of *Alalili* systems was recorded under the communal category which comprised 82% compared to 18% of *Alalili* systems that were recorded under the private category. Communal *Alalili* systems generally depicted the highest spatial distribution across land uses compared to that of private *Alalili* (Fig 6).

**3.3.2 Overall *Alalili* abundance and distribution across land use and class-size.**   The abundance of *Alalili* varied significantly ($\chi^2 = 39.2$, df = 3, $p < 0.001$) from GCA to other land use categories whereby, 50% of *Alalili* systems were recorded from the GCA. The NCA had an

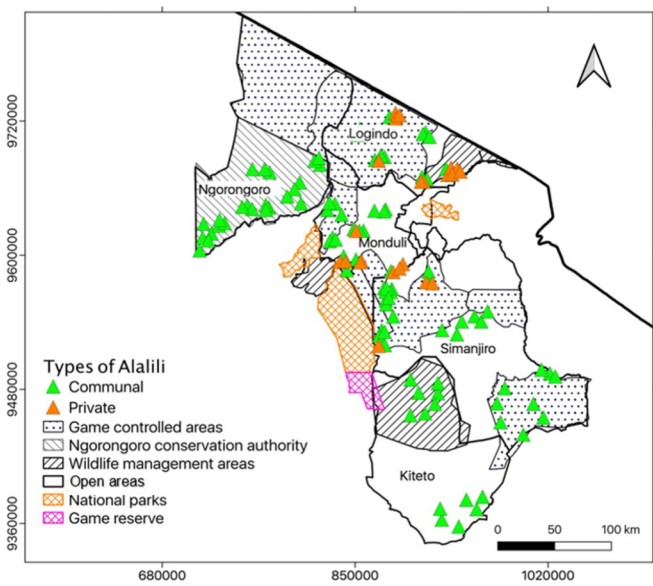

**Fig 6. General distribution of the surveyed *Alalili* types across rangeland areas of Northern Tanzania.**

overall abundance of 20% *Alalili* systems, open areas had 16% of *Alalili* systems, and WMA had 14% of *Alalili* systems. For each land use category, *Alalili* distribution varied with regard to class sizes whereby GCA had 51%, 37%, and 12% of small, large, and medium-sized *Alalili* systems respectively. NCA had 83.3%, 8.3%, and 8.3% of small, large, and medium-sized *Alalili* systems respectively while open areas had 74%, 16%, and 10% of small, medium, and large-sized *Alalili* systems respectively. WMA on the other hand had 76.5% and 23.5% of small and large-sized *Alalili* systems respectively and none medium-sized *Alalili* systems (Fig 7).

**3.3.3 Abundance and distribution of communal *Alalili* systems across land use and size categories.** The highest abundance of communal *Alalili* systems was recorded in the GCA with a total proportion of 48% *Alalili* followed by NCA with an abundance proportion of 25% communal *Alalili* systems. The open areas comprised of 17% communal *Alalili* while WMA had the least abundance of 10% communal *Alalili* systems. An abundance within each land use depicted different distribution patterns of communal *Alalili* systems. GCA had 53.2%, 34.0%, and 12.8% of small, large, and medium-sized *Alalili* systems respectively. NCA had 83.3%, 8.3%, and 8.3% of small, large, and medium-sized *Alalili* respectively. Open areas were featured by 81.3%, 12.5%, and 6.2% of small, medium, and large-sized *Alalili* systems respectively. WMA on the other hand comprised 70% and 30% of small and large *Alalili* systems respectively and none of them was medium sized (Table 2).

Based on size categories, small size recorded the highest abundance of communal *Alalili* which was 67% followed by large size which comprised 23% of *Alalili* systems. Medium size recorded the least abundance of communal *Alalili* encompassing 10% of *Alalili* systems (Table 2). Further distribution analysis within each size category depicted that 38.5%, 30.8%, 20%, and 10.7% of small-sized *Alalili* systems are found in GCA, NCA, open areas, and WMA respectively. 72.7%, 13.6%, 9.1%, and 4.6% of large-sized *Alalili* systems are located in the GCA, WMA, NCA, and in open areas respectively. There were 60%, 20%, and 20% of medium-sized *Alalili* systems in GCA, NCA, and in the open areas respectively, and none in WMA (Fig 8*A*).

**3.3.4 Abundance and distribution of private *Alalili* systems across land use and size categories.** GCA recorded the highest abundance of private *Alalili* systems which was 54%

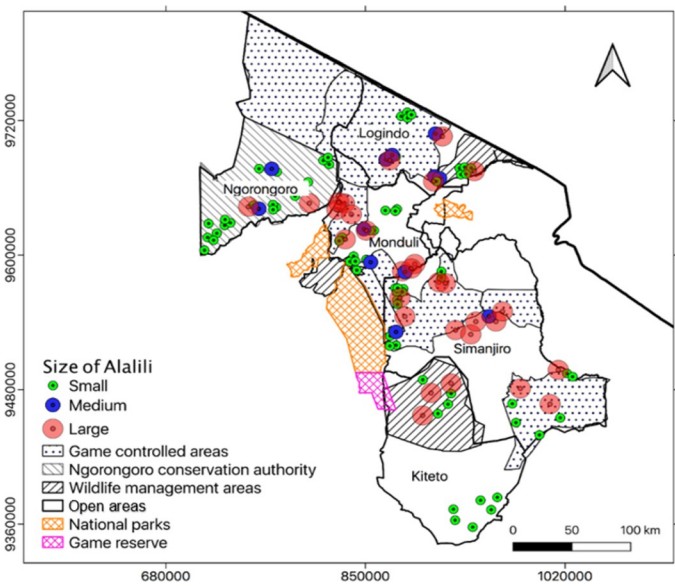

**Fig 7. Generalized distribution of the surveyed *Alalili* systems by size across land use categories.**

**Table 2. Summary of *Alalili* abundance across types, land use, and size categories.**

| Land use | Type of *Alalili* | Abundance by Class size | | | Total |
|---|---|---|---|---|---|
| | | **Large** | **Medium** | **Small** | |
| GCA | Communal | 16 | 6 | 25 | 47 |
| | Private | 6 | 1 | 5 | 12 |
| NCA | Communal | 2 | 2 | 20 | 24 |
| | Private | 0 | 0 | 0 | 0 |
| Open Areas | Communal | 1 | 2 | 13 | 16 |
| | Private | 1 | 1 | 1 | 3 |
| WMA | Communal | 3 | 0 | 7 | 10 |
| | Private | 1 | 0 | 6 | 7 |
| Total | Communal | 22 | 10 | 65 | 97 |
| | Private | 8 | 2 | 12 | 22 |
| | Total | 30 | 12 | 77 | 119 |

*Abbreviations: GCA, Game Controlled areas; NCA Ngorongoro Conservation Area; WMA, Wildlife Management Area.

followed by WMA which comprised 32% of *Alalili* systems. Open areas recorded the least abundance with 14% of private *Alalili* systems (Table 2). In contrast to communal *Alalili*, private *Alalili* systems were not observed in the NCA. Further analysis within each land use depicted different distribution pattern with regard to class sizes. In GCA, there were 50%, 42%, and 8% of large, small, and medium-sized *Alalili* systems respectively. In WMA, there were 86% and 14% of small and large sized *Alalili* respectively and none was recorded under medium-sized category. In open areas, each category of the size comprised of 33.3% of private *Alalili* system (Fig 8*B*).

On the other hand, small size recorded the highest abundance of private *Alalili* which covered 55% followed by large size which comprised 36% of private *Alalili* systems. Medium size recorded the least abundance encompassing 9% of private *Alalili* systems. The distribution analysis of private *Alalili* within each size category was done with regard to land use where by 50%, 42%, and 8% of small-sized private *Alalili* systems were found in WMA, GCA, and open areas respectively. Moreover, 75%, 12.5%, and 12.5% of large-sized private *Alalili* systems were located in GCA, WMA, and open areas respectively. There was 50% of medium-sized private *Alalili* systems in both GCA and in open areas while none was recorded in WMA. NCA had no private *Alalili* silvo-pastoral conservation systems (Table 2).

### 3.4 Effects of land uses and *Alalili* types on the size of *Alalili* systems

There was a significant negative association of *Alalili* size between *Alalili* systems in the GCA and those in the NCA (β = -1081.51, S.E. = 183.00, *p<0.001*). But the association of *Alalili* size for GCA was not significant to that of both open areas (β = -292.47, S.E. = 172.53, *p = 0.0900*) and WMA (β = 53.45, S.E. = 293.42, *p = 0.8555*) (Table 3).

Furthermore, a significant negative association of *Alalili* size between communal and private *Alalili* systems was observed (β = -1352.10, S.E. = 361.34, *p<0.001*). On the other hand, there was a significant negative association of *Alalili* size between large-sized *Alalili* systems and those of both small-sized *Alalili* systems (β = -2538.84, S.E. = 639.66, *p<0.001*) and medium-sized *Alalili* systems (β = -1795.52, S.E. = 615.46, *p = 0.0035*; Table 3).

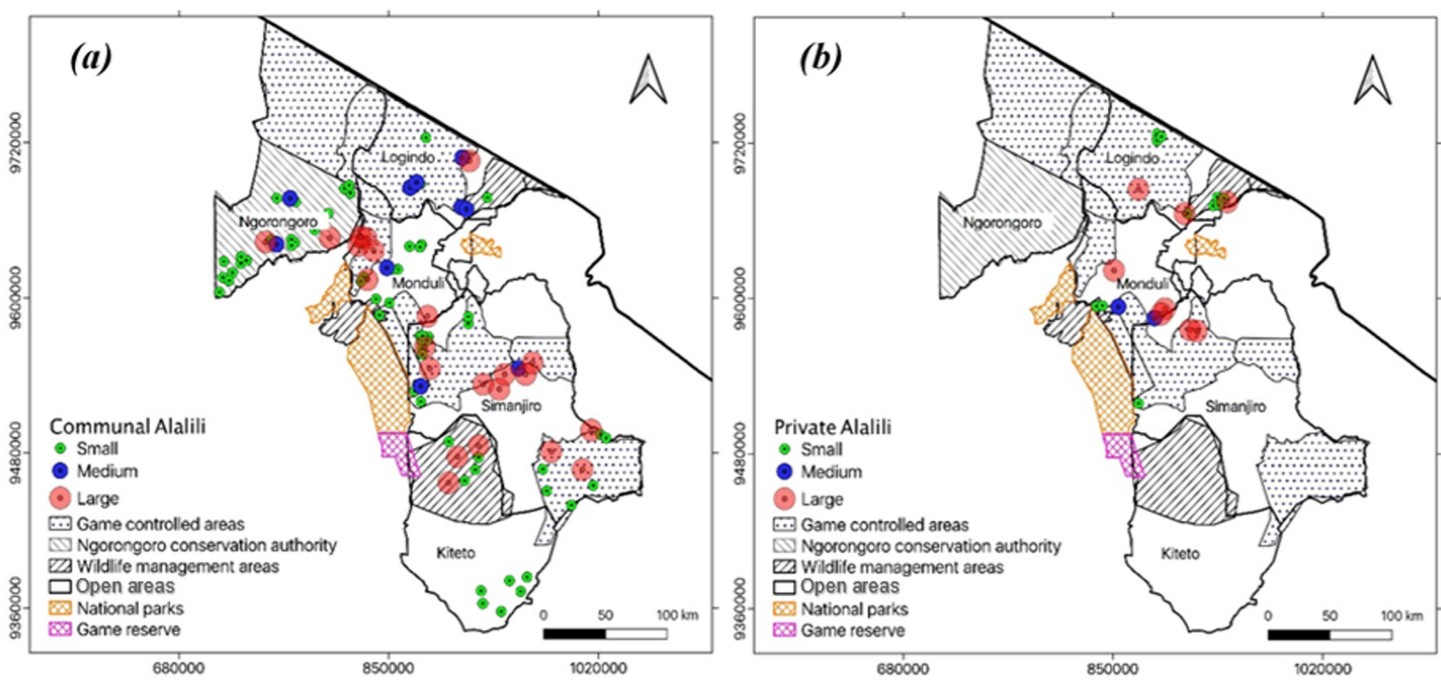

**Fig 8.** Distribution of *Alalili* systems across land use and size: *(a) Communal (b) Private.*

## 3.5 Correlation effects of *Alalili* size and age on livestock herd size

A generalized estimation equation model revealed that the livestock herd size was positively correlated to both size and age of *Alalili* systems and the association was significant across the *Alalili* size but not significant along the age (Table 4). The livestock herd size was higher in the private *Alalili* systems compared to the reference category communal *Alalili* systems depicting a significant negative association between them (Table 4). On the other hand, the livestock herd size was higher and lower in the *Alalili* systems of the open areas and NCA respectively compared to *Alalili* systems of reference category GCA depicting a negative association. While the relationship of livestock herd size between *Alalili* systems in the NCA and GCA was

**Table 3. Summary statistics of generalized estimation equation (GEE) model explaining associations between explanatory variables and the size of *Alalili* systems in the study area.** Land use, class size, and type of *Alalili* were defined as factors whereby the baseline variables are GCA, large size, and communal *Alalili* respectively.

| | Estimate | Std. Error | Wald | Pr(>\|W\|) |
|---|---|---|---|---|
| (Intercept) | 3115.63 | 636.90 | 23.930 | < 0.001 *** |
| Land use-NCA | -1081.51 | 183.00 | 34.925 | < 0.001 *** |
| Land use-OA | -292.47 | 172.53 | 2.874 | 0.0900 |
| Land use-WMA | 53.45 | 293.42 | 0.033 | 0.8555 |
| Types of *Alalili*-Private | -1352.10 | 361.34 | 14.002 | < 0.001 *** |
| Size-Medium | -1795.52 | 615.46 | 8.511 | 0.0035 ** |
| Size-Small | -2538.84 | 639.66 | 15.754 | < 0.001 *** |

Significant codes:

* $p < .05$

** $p < .01$

*** $p < .001$. Abbreviation: GCA, Game Controlled Areas; NCA Ngorongoro Conservation Area; OA, Open Areas; WMA, Wildlife Management Area.

**Table 4. Summary statistics of generalized estimation equation (GEE) model explaining associations between explanatory variables and the number of livestock in the study area.** Land use and type of *Alalili* were defined as factors whereby the baseline variables are GCA and communal *Alalili* respectively.

| | Estimate | Std. Error | Wald | Pr(>|W|) |
|---|---|---|---|---|
| (Intercept) | 551.37 | 182.96 | 9.080 | 0.0026 ** |
| Area of *Alalili* | 1.39 | 0.04 | 996.150 | < 0.001*** |
| Age of *Alalili* | 1.18 | 2.82 | 0.180 | 0.6741 |
| Type of *Alalili* Private | -402.39 | 121.22 | 11.020 | < 0.001*** |
| Land use NCA | -377.60 | 107.05 | 12.440 | < 0.001*** |
| Land use OA | -142.09 | 86.79 | 2.680 | 0.1016 |
| Land use WMA | 223.69 | 206.25 | 1.180 | 0.2781 |

Significant codes:

* *p < .05*

** *p < .01*

*** *p < .001*. Abbreviation: GCA, Game Controlled Areas; NCA Ngorongoro Conservation Area; OA, Open Areas; WMA, Wildlife Management Area.

significant, the association between *Alalili* systems in the open areas and GCA was not significant (Table 4). Although *Alalili* systems in WMA depicted a higher livestock herd size compared to *Alalili* systems in the reference category GCA, the association was not significant (Table 4).

## 4. Discussion

Indigenous local knowledge (ILK) and research around it has emerged as an important area of sustainability science, informing and supporting pressing challenges of our modern times. It provides information and data that can increase resilience, promote sustainable restoration and rewilding for transforming communities in landscapes towards more resilient, sustainable, and equitable futures [8, 75, 76]. Many rangeland areas harbor communities with different socio-cultural practices most of which are not documented [16]. Due to lack of recognition of these, rangelands and their indigenous socio-cultural practices are rarely going to survive the current anthropogenic threats. This study was able to distinctively characterize *Maasai Alalili* by land use types, separating private and communal ones into three size categories named small, medium, and large *Alalili* systems. The documentation of this which had not been reported in the previous studies [5, 12, 17] are potential tools for facilitating management initiatives of the traditional fodder conservation systems. Through predicting the possible loss of *Alalili* systems, the majority of these being communal with reference made to private that vanished in non-protected land, their future practices may be prone to total loss and extinction.

### 4.1 Status of *Alalili* systems and the suggested changes across types

On average this study identified and categorized more communal *Alalili* (82%) than private (18%) throughout the studied rangeland area predicting that this practice at the household level is being threatened by various drivers that may include land use and socio-cultural changes [12, 18, 25, 50]. Although, the *Maasai* pastoral communities in northern Tanzania are reported to practice communal ownership of rangelands as the dominant system [18, 44], this work suggests the historical evolvement of private *Alalili* systems that was not documented for many years as supported by recent works [17, 18, 77, 78]. This is further supported by a few surveyed old-aged private *Alalili* systems reported to historically exist since the past 40 to 70 years as informed by the rapporteur in the focused group discussion;

**FGD-MOND-22:** "*This Alalili system is privately owned by our clan and has existed for many years. Our grandparent narrated a brief history about it, whereby they have been keeping it since the colonial era in the late 1950s. Since that time, members of the clan have been using it not only for grazing purposes, but also for conducting rituals, cursing practices over the deviants from management norms of Alalili that built fear over our livestock herds and respect among the surrounding households. Some neighbour households also had their Alalili systems, but they did not manage to uphold them due to an introduction of the villagelization and communism policies in the 1970s which made them remain dependent on the communal grazing areas*" (FGD/*Alalili* survey/Loosimingori village/June, 2022).

**FGD-LONG-24:** "*The large portion of this private Alalili that existed since 1970s, has been very important site of securing pastures that sustain our livestock especially the calves and lactating cattle in the long dry seasons. However, due to increased number of household members, we have changed a little portion (especially the eastern side you see) for residential purposes where my two sons have established their families. Due to a reduced grazing area, one of my grandsons is currently engaged in the mining activities at the nearby village of Matale A to sustain family needs. Some other households in the neighbourhoods have sold their privately owned grazing areas to immigrants who practice crop farming (maize, beans and sunflower cultivation)*" (FGD/*Alalili* survey/Kitumbeine village/May, 2022).

Having an 18% of private *Alalili* systems, that are very recently established, is regarded as a newly emerging practice over the historical communal ownership of grazing lands by *Maasai* pastoralists [13, 17]. Although, from an opposite community perspective, it is regarded as a crucial restoration initiative whereby the *Maasai* pastoralists are securing the grazing areas that are likely being changed into protected areas, farmland, and settlements [18, 78]. They are turning into private ownership as a way to sustain their historical indigenous and traditional way of conserving pasture lands that used to support their grazing activities and rangeland management [17].

The changes are observed to be in transition in *Maasai* culture, eroding the key livelihood practices that is vital for the survival of these communities and sustainable use of the biodiversity [18]. Similarly, most of the private *Alalili* are smaller in size making them even more prone to vanish due to higher livestock herd size and grazing intensity compared to that of the communal *Alalili* systems [17, 25]. The findings suggest that communal *Alalili* systems are currently holding significant conservation potential than private *Alalili*. Regardless of the recent establishment of private *Alalili* systems, yet there is a radical shift of conservation priority from these private to communal areas because, conservation agencies are highly promoting it through conservation incentives provided to community-based organizations for climate change mitigation [46, 78]. In the past three decades, numerous projects have been done for rangeland improvement through participatory rangelands management (PRM) exposing more conservation incentives to communal enclosures than in private [26, 79].

Additionally, policies for livestock, wildlife, tourism, and land sectors in Tanzania have prioritized measures that promote communal rangeland management through traditional land use practices [26, 50, 56–58, 63, 80]. Effects of poverty, climate change, and food insecurity are additional drivers that scale down the size of private *Alalili* systems, abundance, and distribution [7, 71, 81]. The growing population (both human and livestock) are in need of expanding settlement areas and farmlands for food crop production within the same land resources leading to competitive effect and degradation [71]. A portion of privately reserved pasture land is sold out for sustaining their livelihood [8, 82]. Thus, private *Alalili* systems are at risk of malpractices compared to communal ones. On the other hand, the decreasing number and size of

private *Alalili* is notifying rangeland managers and experts on the communal *Alalili* systems that are predicted to face a risk of collapse with regard to the theory of "*tragedy of the commons*" [83, 84]. Efforts therefore should be directed towards increasing the number and size of private *Alalili* as well as their practices outside protected areas to ensure sustainable fodder and or pasture availability at private pasture land [6, 7, 85]. The existence of both communal and private *Alalili* would strengthen a sense of rangeland conservation advocacy among *Maasai* pastoralists who maintain their cultural heritage with minimum harmful threats to biodiversity resources [5, 18, 44].

The observed transition from pastoralism to crop cultivation among *Maasai* pastoral communities has intensified losses in pasture resources with regard to reduced grazing land size and rangeland encroachment with less return on crops [45, 86]. Promoting *Alalili* would bring about ten times return when sustainable grazing is practiced [13, 17, 18, 23]. Utilization of *Alalili* systems do not only serve as useful in-situ fodder/forage banks but also useful drivers in reducing impacts of overgrazing and rangeland degradation for upgrading the health of biodiversity and its associated ecosystem services [5, 13, 48]. They play multiple roles such as feed reserves, environmental and wildlife conservation, economic development, mitigating impacts of global climate change including soil erosion, invasive species, and bush encroachment as well as giving ecosystem services [1, 4, 17, 78, 79].

## 4.2 Occurrence of *Alalili* systems across land use categories

We found that more than half of all *Alalili* systems in the rangelands of northern Tanzania are found in GCA. This trend suggests that, apart from the largest proportion of land surface occupied by GCA (45%) above NCA (20%), open areas (19%), and WMA (16%), promotion of this practice within the GCA might be the reason for this. Other works report that, being community-based initiatives for protecting wildlife from degradation and fragmentation, the formal institution of GCA and WMA are responsible for administering the limit in accessing the extreme dry season grazing areas [15, 78]. In the NCA there were no private *Alalili* and that the ownership of land is devoted to the Ngorongoro conservation area authority and not villagers [12, 45]. In the open areas, there was no significant size variation between private and communal *Alalili* systems suggesting that they face similar encroachment pressures from villagers and village governments [80]. Portions of communal and some private enclosures are changed to potential infrastructural areas where schools, health care centers, markets, and food stores are constructed to support services such as education, health, and other basic necessities [18]. This suggests that LULCC due to the cultural and traditional shift from grazing to crop cultivation among the *Maasai* society, especially in unprotected areas, poses a high pressure on traditionally conserved pastures leading to their loss [45, 46, 87]. Overall, the potential of *Alalili* to act as a restoration pathway in the degraded rangeland patches can be a strategy with the potential to be up scaled particularly when the benefits of this practice are reiterated to the community with a reflection made from *Ngitili* systems which are multi-functional ecosystems [5, 7, 16, 48]. This requires a need for the range map results on the status of *Alalili* types regarding their size, abundance, and distribution be communicated back to communities so they know their extent of disappearance as a wakeup call for them to take measures and action to restore this in their land.

We emphasize the need for creating pastoral groups through community champions to promote the intensification of *Alalili* practices through incentive provisioning such as payments and rewards for ecosystem services, green loans, REDD+, and subsidies [88]. Communities should be proactively engaged and informed about the need to protect this cultural practice by describing the wisdom of ILK on rangeland management associating its values and

benefits to avoid its collapse in the community. In promoting this practice to the household level, models that aim at managing rangelands and pasture should integrate this ILK and the economic, social, and environmental dimensions that are mutually dependent on one another [6]. The areas of implementation priorities should be known private lands where members of the households can easily be embraced when managing their pastureland. Furthermore, species that are native and palatable to livestock should be restored in the degraded *Alalili* as an incentive in addition to formulation and implementation of community by-laws that support its sustainable intensification.

## 5. Conclusion

Approaches that are developed for managing rangelands in northern Tanzania are not necessarily developed based on indigenous cultural heritage such as *Alalili*, risking their success and sustainability [89]. Recognizing the *Alalili* in such approaches minimizes the risk of misjudging their importance by outweighing the benefits and values that come up with the inclusion of *Alalili*, resulting in the wide-community level adoption and dissemination [26]. In contrast to modern natural science outputs, the presented results have indicated that *Maasai Alalili* silvo-pastoral conservation systems are still in existence but their sustainability is highly at risk, especially in private land areas. The generated results could facilitate the inclusion of their practices into core pasture production and management areas, facilitating their reinforcement into policy and practices. We encourage the extended use of our survey results in further assessing the biomass and other compliance practices in screening the best management practices when implementing *Alalili* in the landscape level. However, their size seemed to be decreasing signaling their possible loss in the near future, thus necessitating on a temporal analysis to determine the degree of sustainability risk. The stakeholders and communities should be informed about emerging effects of LULCC, growing population and environmental pressures such as climate change that threaten the sustainability of traditional conservation practices including the *Maasai Alalili* systems. Though the primary concern of establishing *Maasai Alalili* systems was to preserve pastures during the hardship times of pasture availability, the pastoral communities in rangeland areas need to be informed about extended benefits of *Alalili* systems. Despite the role played by private *Alalili* systems to the livestock and wildlife, they are faced with a confounding threat to deteriorate compared to communal *Alalili*. Further loss of *Alalili* conservation systems will not only lower down availability of pasture resources, but will also discourage initiatives for restoration of degraded rangelands through recovery of vegetation. Based on our findings, initiatives aimed at improving *Alalili* systems conservation and management strategies need to be considered. We recommend further assessment of the existing *Alalili* systems management typologies fitting to utilization compliance and that they should be made formal. Such results highlight the need for incorporation of ILK in grassland conservation and management policies allowing and instructing on the call for frequency and extent of *Alalili* practices as a management tool for both rangelands and their community resilience in East African rangelands.

## Supporting information

**S1 Appendix. *Alalili* compiled data.**
(XLSX)

## Acknowledgments

We owe our special gratitude to the Nelson Mandela African Institution of Science and Technology for granting the permission to conduct this research work. We are also very much indebted for the permit (2022-222-NA-2022-005 dated 20th April, 2022) we got from COST-ECH, TAWIRI, TAWA, NCAA, and Local government authorities of each district for allowing access to *Alalili* systems within rangelands throughout the study area. We would also like to acknowledge the field and support team - Mr. Emmanuel Mboya (Tanzania Plant Health and Pesticides Authority-TPHPA), Mr. John Erasto Sanare (Tanzania Wildlife Research Institute), Mr. Neovitus Siang'a and Mr. Kirerenjo Mereso (Tanzania People and Wildlife), Ms. Catherine Maembe, Mr. Emmanuel Lorru, Mr. Nganana M. Papalay and Mr. Lomayani Lukumay (District Game Officers), Mr. Birikaa R. Olesikilal, Mr. Danstan Mndolwa and Mr. Mamus Toima (District Rangeland Officers).

## Author Contributions

**Conceptualization:** Elkana Hezron, Issakwisa B. Ngondya, Linus K. Munishi.

**Data curation:** Elkana Hezron.

**Formal analysis:** Elkana Hezron.

**Funding acquisition:** Elkana Hezron.

**Investigation:** Elkana Hezron, Issakwisa B. Ngondya, Linus K. Munishi.

**Methodology:** Elkana Hezron, Issakwisa B. Ngondya, Linus K. Munishi.

**Project administration:** Elkana Hezron, Issakwisa B. Ngondya, Linus K. Munishi.

**Resources:** Elkana Hezron, Issakwisa B. Ngondya, Linus K. Munishi.

**Software:** Elkana Hezron.

**Supervision:** Issakwisa B. Ngondya, Linus K. Munishi.

**Validation:** Elkana Hezron, Issakwisa B. Ngondya, Linus K. Munishi.

**Visualization:** Elkana Hezron, Issakwisa B. Ngondya, Linus K. Munishi.

**Writing – original draft:** Elkana Hezron, Issakwisa B. Ngondya, Linus K. Munishi.

**Writing – review & editing:** Elkana Hezron, Issakwisa B. Ngondya, Linus K. Munishi.

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
