## [Decision Letter · Decision Letter 0]

9 Jan 2024

PONE-D-23-34899Sustaining indigenous Maasai Alalili silvo-pastoral conservation systems for improved community livelihood and biodiversity conservation in East African rangelandsPLOS ONE

Dear Dr. Hezron,

Thank you for submitting your manuscript to PLOS ONE. After careful consideration, we feel that it has merit but does not fully meet PLOS ONE’s publication criteria as it currently stands. Therefore, we invite you to submit a revised version of the manuscript that addresses the points raised during the review process.

Please pay paprticular attention to the attached document with reviewer's comments. Please submit your revised manuscript by Feb 23 2024 11:59PM. If you will need more time than this to complete your revisions, please reply to this message or contact the journal office at plosone@plos.org. Please include the following items when submitting your revised manuscript:A rebuttal letter that responds to each point raised by the academic editor and reviewer(s). You should upload this letter as a separate file labeled 'Response to Reviewers'.A marked-up copy of your manuscript that highlights changes made to the original version. You should upload this as a separate file labeled 'Revised Manuscript with Track Changes'.An unmarked version of your revised paper without tracked changes. You should upload this as a separate file labeled 'Manuscript'.

We look forward to receiving your revised manuscript.

Kind regards,

Luca Nelli, PhD

Academic Editor

PLOS ONE

Journal Requirements:

2. Thank you for submitting the above manuscript to PLOS ONE. During our internal evaluation of the manuscript, we found significant text overlap between your submission and previous work in the [introduction, conclusion, etc.].

Please revise the manuscript to rephrase the duplicated text, cite your sources, and provide details as to how the current manuscript advances on previous work. Please note that further consideration is dependent on the submission of a manuscript that addresses these concerns about the overlap in text with published work.

[If the overlap is with the authors’ own works: Moreover, upon submission, authors must confirm that the manuscript, or any related manuscript, is not currently under consideration or accepted elsewhere. If related work has been submitted to PLOS ONE or elsewhere, authors must include a copy with the submitted article. Reviewers will be asked to comment on the overlap between related submissions (http://journals.plos.org/plosone/s/submission-guidelines#loc-related-manuscripts).]

We will carefully review your manuscript upon resubmission and further consideration of the manuscript is dependent on the text overlap being addressed in full. Please ensure that your revision is thorough as failure to address the concerns to our satisfaction may result in your submission not being considered further.

"This work was funded by the Nelson Mandela African Institution of Science and Technology through African Development Bank (AfDB) (Grant No: P-Z1-IA0-016), Higher Education for Economic Transformation (HEET) project (Grant No: IDA68870) as well as the Rufford small research grants (Grant No: 37388-1).  "

5. We note that your Data Availability Statement is currently as follows: [All relevant data are within the manuscript and its Supporting Information files.]

7. We note that [Figures 1, 6, 7 and 8] in your submission contain [map/satellite] images which may be copyrighted. All PLOS content is published under the Creative Commons Attribution License (CC BY 4.0), which means that the manuscript, images, and Supporting Information files will be freely available online, and any third party is permitted to access, download, copy, distribute, and use these materials in any way, even commercially, with proper attribution. For these reasons, we cannot publish previously copyrighted maps or satellite images created using proprietary data, such as Google software (Google Maps, Street View, and Earth). For more information, see our copyright guidelines: http://journals.plos.org/plosone/s/licenses-and-copyright.

a. You may seek permission from the original copyright holder of Figures 1, 6, 7 and 8 to publish the content specifically under the CC BY 4.0 license.  

Reviewers' comments:

Reviewer's Responses to Questions

**Comments to the Author**

1. Is the manuscript technically sound, and do the data support the conclusions?

Reviewer #1: Yes

Reviewer #2: Partly

2. Has the statistical analysis been performed appropriately and rigorously? 

Reviewer #1: Yes

Reviewer #2: Yes

3. Have the authors made all data underlying the findings in their manuscript fully available?

Reviewer #1: Yes

Reviewer #2: No

4. Is the manuscript presented in an intelligible fashion and written in standard English?

Reviewer #1: Yes

Reviewer #2: Yes

5. Review Comments to the Author

Reviewer #1: PONE-D-23-34899-Sustaining Alalili systems in rangelands of East Africa

The research paper employs a cross-sectional research design to investigate the existing Maasai Alalili silvo-pastoral conservation systems in East African rangelands. The findings reveal that while these systems are still present, their sustainability is under significant threat.

Here are some suggestions for enhancing the clarity and robustness of the paper:

1. Address the lack of supporting evidence for the claim about decreasing Alalili sizes by including graphics, analyses, or references. Consider exploring the relationship between Alalili size and livestock herd size. Additionally, expand the literature review to encompass a similar system in southern Kenya known as "Olopololi" (refer to https://amboseliecosystem.org/our-campaigns/#:~:text=01.-,Olopololi%20Plots,-The%20campaign%20is).

2. Provide exact p-values for all values greater than or equal to 0.001, adhering to PLOS ONE Submission guidelines.

3. Enhance figure captions by incorporating additional text to clarify the figure contents.

4. Cite QGIS and other software/tools used in the analysis.

5. Improve the clarity of the formulation and interpretation of the Generalized Estimation Equation (GEE) model in Table 3. Clearly define the response and explanatory variables, specify categories used as references, and offer detailed model information in the supplementary material to assist readers in understanding the analysis.

Reviewer #2: The manuscript describes the indigenous Maasai Alalili silvo-pastoral conservation systems of livestock keeping, which can best be described as a traditional, rotational grazing patterns or a modern fodder bank systems where a large portion of land is reserved for dry spells or drought. These traditional systems have been successful in the past but current socioeconomic and political factors threatened their continued existence. Hence, there is a need to understand how these systems are currently categorised across livestock keeping communities in northern Tanzania, and how best they can be sustained. I have provided an attachment containing detail suggestions on how the manuscript can be improved.

6. PLOS authors have the option to publish the peer review history of their article (what does this mean?). If published, this will include your full peer review and any attached files.

Reviewer #1: **Yes: **Victor N. Mose

Reviewer #2: No

---

## [Author Response · Author response to Decision Letter 0]

12 Feb 2024

We are grateful for the useful observation and comments provided by both Editors and Reviewers. We would like to confirm that, through this submission, all the comments that were raised by Editors and reviewers are addressed as indicated in the rebuttal letter named "Response to Reviewers".

---

## [Decision Letter · Decision Letter 1]

3 Apr 2024

PONE-D-23-34899R1Sustaining indigenous Maasai Alalili silvo-pastoral conservation systems for improved community livelihood and biodiversity conservation in East African rangelandsPLOS ONE

Dear Dr. Hezron,

Thank you for submitting your manuscript to PLOS ONE. After careful consideration, we feel that it has merit but does not fully meet PLOS ONE’s publication criteria as it currently stands. Therefore, we invite you to submit a revised version of the manuscript that addresses the points raised during the review process.

Please make sure to address the final concerns of reviewer #1.

We look forward to receiving your revised manuscript.

Kind regards,

Luca Nelli, PhD

Academic Editor

PLOS ONE

Journal Requirements:

Reviewers' comments:

Reviewer's Responses to Questions

**Comments to the Author**

1. If the authors have adequately addressed your comments raised in a previous round of review and you feel that this manuscript is now acceptable for publication, you may indicate that here to bypass the “Comments to the Author” section, enter your conflict of interest statement in the “Confidential to Editor” section, and submit your "Accept" recommendation.

Reviewer #1: All comments have been addressed

2. Is the manuscript technically sound, and do the data support the conclusions?

Reviewer #1: Yes

3. Has the statistical analysis been performed appropriately and rigorously? 

Reviewer #1: Yes

4. Have the authors made all data underlying the findings in their manuscript fully available?

Reviewer #1: Yes

5. Is the manuscript presented in an intelligible fashion and written in standard English?

Reviewer #1: Yes

6. Review Comments to the Author

Reviewer #1: For Table 3 and 4 include precise p-values for all values greater than or equal to 0.001, in line with PLOS ONE Submission guidelines. For smaller values “P < 0.0001” is neater.

A careful reading of the manuscript is necessary to ensure that all in-text citations are properly referenced, including the tools and software used.

Regarding the GEE model, the correlation structure is specified as "independent," suggesting no correlation between responses within subjects. Is this assumption accurate? Given the potential correlation among explanatory variables, the correlation structure "exchangeable" may be more suitable.

7. PLOS authors have the option to publish the peer review history of their article (what does this mean?). If published, this will include your full peer review and any attached files.

Reviewer #1: **Yes: **Victor N. Mose

---

## [Author Response · Author response to Decision Letter 1]

23 Apr 2024

We have addressed all the comments that were raised by reviewers as indicated in the attached rebuttal letter and the revised manuscript.

---

## [Editor Report · Decision Letter 2]

30 Apr 2024

Sustaining indigenous Maasai Alalili silvo-pastoral conservation systems for improved community livelihood and biodiversity conservation in East African rangelands

PONE-D-23-34899R2

Dear Dr. Hezron,

We’re pleased to inform you that your manuscript has been judged scientifically suitable for publication and will be formally accepted for publication once it meets all outstanding technical requirements.

Kind regards,

Luca Nelli, PhD

Academic Editor

PLOS ONE
---

## [Editor Report · Acceptance letter]

4 May 2024

PONE-D-23-34899R2 

PLOS ONE

Dear Dr. Hezron, 

I'm pleased to inform you that your manuscript has been deemed suitable for publication in PLOS ONE. Congratulations! Your manuscript is now being handed over to our production team.

Kind regards, 

on behalf of

Dr. Luca Nelli 

Academic Editor

PLOS ONE